# Production of Particleboard Using Various Particle Size Hemp Shives as Filler

**DOI:** 10.3390/ma15030886

**Published:** 2022-01-24

**Authors:** Kristaps Zvirgzds, Edgars Kirilovs, Silvija Kukle, Uldis Gross

**Affiliations:** 1Institute of Design Technologies, Faculty of Materials Science and Applied Chemistry, Riga Technical University, LV-1658 Riga, Latvia; edgars.kirilovs@rtu.lv (E.K.); silvija.kukle@rtu.lv (S.K.); 2Department of Information Technologies, Latvia University of Life Sciences and Technologies, LV-3001 Jelgava, Latvia; fkgross@llu.lv

**Keywords:** hemp shives, renewable resources, particleboard, particle size, formaldehyde resin glue

## Abstract

Research was performed into the use of hemp shive as a fast-growing and carbon-storing agricultural waste material in the production of particleboard for the construction industry. Hemp shives were acquired and prepared for board production with the use of milling and sieving to reach two target groups with 0.5 mm to 2 mm and 2 mm to 5.6 mm particle size ranges. The cold pressing method was used to produce hemp boards with Kleiberit urea formaldehyde resin as a binder. The boards were made as 19 mm thick single-layer parts with a density range of 300 ± 30 kg/m^3^, which qualifies them as low-density boards. Exploratory samples were made using milled hemp fibers with higher density. Additional components such as color pigments and wood finishes were added to test improved features over raw board samples. Tests were performed to determine moisture contents, density range, structural properties, and water absorption amounts. Produced board bending strength reached 2.4 MPa for the coarser particle group and thermal conductivity of 0.057 ± 0.002 W/(mK). The results were compared with existing materials used in the industry or in the development stage to indicate options of developed board applications as indoor insulation material in the construction industry.

## 1. Introduction

In the modern-day world, the demand for wood fiber products for a growing population and competing industries is contrasted by a decline of sustainable timber resources. Both the demand for material quality and the need for more environmentally friendly technologies are increasing. Institutions such as the United Nations and the European Union are making agreements such as the Paris Agreement in 2015 or the Glasgow conference in 2021 to increase the demand for ecological technologies and materials so that the ecological footprint is reduced to zero [1,2,3].

In the construction and manufacturing industries, there are two main areas that could help reach the objectives set in these agreements—research into the optimization of existing, new technologies and the development of new materials. As of 2021, the existing applications need new approaches in both areas. Research into the identification of energy loss, life cycle assessment for materials, improved durability standards, and the switch to more carbon-neutral materials is already in progress. [3,4,5,6,7,8,9]. The use of plant-based ingredients and green biomass is an imminent alternative to timber resources and soon will be explored even more than in the past [10,11,12]. In some cases, plant biomass is advantageous to use because of its low production cost, fast growth, and self-sustaining character due to the quicker renewability of their source. The speed of their growth shows very promising results from an environmental impact and commercial point of view if the proper technological application is utilized [10,13,14,15,16,17].

From all the cultivated plants, hemp is chosen as a filler material because it is one of the most sustainable of fast-growing agricultural crops. It can be harvested twice a year, so it is twice as effective in reducing the greenhouse effect. During its growth, hemp is a crop with a very high CO_2_ sequestration amount—industrial hemp absorbs between 8 and 15 tons of CO_2_ per hectare of cultivation. In comparison, forests typically capture 2 to 6 tons of CO_2_ per hectare per year depending on the number of years of growth, the climatic region, and the type of trees [18]. Numerous studies estimate that hemp is one of the best CO_2_-to-biomass converters. In many agricultural holdings that cultivate hemp for the food industry, the shives or even the whole bast, including the fibers, are treated as agricultural waste material and are plowed into the soil; therefore, if properly gathered, it is a raw source material with characteristics similar to wood chips, that could be used to produce composite or board material for construction such as particleboard [8,15,19,20,21]. In 2018, a total of 97 million m^3^ of particleboard was produced worldwide [22].

A lot of research has already been conducted in this field evaluating the options to use hemp shives or other types of plant straws, chips with binders such as lime or cement [16,17,23,24,25,26,27], with real applications already present in the industry. In the Baltics region alone, 10 hemp concrete single-family houses are built each year, according to data from a local construction company that specializes in such buildings. The research of material in such buildings states that CO_2_ sequestrated during the growth time of hemp neutralizes the CO_2_ amount produced in the manufacturing stage, making it a carbon-neutral material [28]. In 2021, for the first time, hemp concrete was used as the main construction material for a public building in France (Pierre Chevet Sports Centre). The use of such material removes the need for additional insulation due to its natural insulating qualities.

More recently, options to use resin or natural starch-type binders have been explored to produce even more environmentally friendly composites and boards from hemp shives, rice husks, sunflower stalk, mycelium, and nut shells [29,30,31,32,33,34,35,36]. Such development is due to rising prices of resources worldwide and the need for more locally supplied stock. Traditionally, particleboard is produced with conventional binders such as melamine-urea formaldehyde (MUF), polycarbonic anhydrides, urea-formaldehyde (UF), and phenol-formaldehyde (PF) [37]. Each has its own hazards to either environment or human health, so a reduction in the binder amount in the final product is important to lessen the negative impact. A plant-based filler can also counter some of the negative aspects, as previously mentioned regarding the hemp crop’s ability to absorb CO_2_.

The type of binders used in similar research is polyurethane resins, which are added in amounts from 1% to 20% of dry mass, most commonly from 8% to 12%. To produce finer results on the outer layers of boards, in one part of a study, a variation of resin amount is used—6–8% in the inner layers and 10–12% in the outer layers. This makes the board smoother, more durable, and more susceptible to coatings and machining [38,39].

When producing a composite material for the manufacturing, woodworking, and building industries, not only the base properties of mechanical strength and possible applications are important, but also additional features such as microbial resistance and thermal conductivity [21,40,41]. Some of these properties, such as microbial resistance, are natural to hemp and cannabinoids, while others have to be enhanced by additional treatments such as alkalization or boiling in water to reduce water absorption amounts and thickness swelling, which usually is high for materials with natural fibers [42,43,44,45].

Hemp has been confirmed as a great filler material in composites for thermal insulation, which is further improved if shives are added to the composite as filler [46]. Hemp shives used as the main ingredient can produce materials with great insulation properties. In previous research, the thermal conductivity values are in the range from 0.064 W/(mK) to 0.074 W/(mK) for hemp shive boards with UF binders or wood fiberboards [47,48]. For hemp shive boards already on the market, the given reference values are in the range from 0.072 W/(mK) to 0.094 W/(mK) [49,50]. In comparison, the most commonly used thermal insulation materials in the construction industry, such as rock wool, glass wool, and phenolic foam, have thermal conductivity values in the range from 0.018 W/(mK) to 0.040 W/(mK) [51].

The aim of this study is to develop lightweight panels manufactured with Latvian hemp shives obtained as a by-product of seed hemp not previously used in the development of high value-added products and evaluate their performance characteristics. The objectives are to (a) test the proposed pressing technology—use of a specialized plywood press consisting of a mold and matrix; (b) assess the chosen particle size and its production technology; (c) identify optimal panel composition without special additives and with special additives to improve their features; (d) produce a sufficient sample that could be used when performing tests to identify panel density, bending strength, water absorption, thermal conductivity, and color equability values.

## 2. Materials and Methods

To produce desired hemp shive board samples, three different components have been chosen with the criterion that they are locally available to limit the environmental impact of transport.

### 2.1. Materials

Experimental mixtures were prepared according to a recipe and consisted of 85% base material (hemp shives), 10% binder, and 5% water), named HS-1, HS-2, and HS-12 in Table 1 (HS—hemp shives) [47]. Additional boards named HS-1A and HA-2A were produced with particle pre-treatment or additive application during particle mixing with binder, described in Section 2.1.3 and Section 2.3, MHF (Milled hemp fiber-shive) boards were produced from leftovers after material processing, described in Section 2.3.

#### 2.1.1. Filler

Hemp (*Cannabis sativa* L.) was grown at the experimental plot of the Agriculture Science Centre of Latgale in Vilani, Latvia, in 2017 [52]. The Polish variety Bialobrzeskie (owner: Institute of Natural Fibres & Medicinal Plants, Poznan, Poland) has been chosen for the experimental part because it is the most common variety of hemp cultivated on eastern Europe plantations [53]. The original particle size distribution of the hemp shives was rather wide (0.063–15 mm), and it contained fine dust particles originating from the manufacturing disintegration process. This material consisted of a large majority of hemp shives and a small amount of hemp bast fibers. The average moisture content of the hemp material was determined by weighing the hemp samples before and after drying them for 24 h at 105 °C according to LVS EN 322 standard and then calculating (Equation (1)) [54].
(1)$$H=\frac{m_{H}-m_{0}}{m_{0}}\times 100\%$$
H—mass of moisture content, %. $m_{H}$—mass of content specimen at normal climate, g. $m_{0}$—mass of specimen after drying, g.

The hemp mass was milled and sieved to form specific particle size groups that were later used in the production of boards (these processes are explained in Section 2.2). A total of 5 hemp shive samples were tested before and after processing.

#### 2.1.2. Binder

For binding purposes, a urea formaldehyde resin (UF) Kleiberit 862.0 (Intarsija Ltd., Riga, Latvia) was used as adhesive to produce the board samples. In the wood product industry, it is mostly used for bonding veneers on chipboard or production of particleboard. This binder can be used to produce E05 (E1) emission class materials according to specifications in EN 717-1, which is the safety standard in Europe. It was chosen because of its physical and chemical properties (UF resins are known for their hardness, low flammability, good thermal properties, and absence of color), cost, and efficiency of use with cold press technology for board materials. This technology could be easily replicated from scientific research to manufacturing processes [47]. The technical data sheet of the binder states that drying time at 20 °C is 4 h and 30 min; at 60 °C it is only 6 min. The weight ratio for the mixture is 100:50 (powder:water) and the open time of the mixed binder is ~5 min.

UF resin has one main disadvantage, the major problem being that it is subject to hydrolytic degradation when in the presence of moisture, water, or acid, so it is expected that produced boards will be water-permeable and can degrade or dissolve quickly when used in an environment with high humidity.

Binder volume in the produced boards in this study is 10%, matching amounts used in similar research [38,39,47]. The application of the binder is described in Section 2.3.

#### 2.1.3. Additives

For improvement of experimental board properties—water resistance, porosity, and visual looks—a group of additives were used—water, oil and alcohol-based colorants, zinc oxide nanopowder, and mineral pigments. See Table 1 for the classification of additive use with specific particle size groups.

The application of additives was made in two ways. The first option explored was pre-treating hemp with the additive in liquid form—mixing it into shives with a construction mixer until a homogeneous mass was formed and then drying it to equilibrium humidity on an open surface. Only then the dried shives were mixed into the molding mass. The second option explored was to add the liquid component during mass mixing for molding simultaneously with the binding agent to reduce the total production time. A possibility of additive application as a coating on an already pressed board surface was explored to compare methods.

### 2.2. Particle Size

The hemp mass was fractionated according to LVS EN 932-5 standard [55]. The sieving was performed by MATEST A059-01 electromagnetic sieve shaker with sieve insert sizes of 5.6 mm, 3.15 mm, 2 mm, 1 mm, 0.5 mm, 0.355 mm, 0.16 mm, and 0.09 mm using an interrupted cycle with a vibration time of 58 s and an interruption of 2 s. The total sieving cycle of each sample lasts 9 min at a vibration intensity of 80 W/m^2^. After the sieving process, the sieves were taken from the machine one by one, each sieve contents were poured into a metal container, the material residues were cleaned from the sieve, and the particles of the respective sieve were weighed.

Fractionization of the supplied hemp shive mass showed that the particles were very coarse—47.6 wt.% of the particles had dimensions exceeding 5.6 mm (Figure 1a). Due to previous research into similar experiments, it was deemed that reduction of hemp shive particle size allows producing more dense and mechanically durable material without a lot of leftovers [10,23,56,57,58]. Possibilities for milling or shredding the hemp mass were considered; one of them—milling with instruments that have blunt edges to reduce hemp fiber twisting—was identified as the best option. A hammer-type cutting mill incorporating sieve inserts of 3.5 mm × 3.5 mm was used to reduce the size of the shives into smaller particles (Figure 1b).

After milling, the mass (Figure 1b) was sieved with the same machinery to evaluate the change in particle size distribution. Particles of size less than 0.5 mm (Figure 1c) were deemed too small and excluded from the study [59]. In the volume of milled shive mass, most particles (85 wt.%) were in the range of 0.5 to 3.15 mm (Figure 1d,e). Particles left on the 5.6 mm sieve consisted of mostly hemp fibers (86 wt.%) and shives (14% wt.%) entangled within fibers (Figure 1f).

Three particle size groups consisting of hemp shives were identified for further studies: from 0.5 mm to 2 mm (Figure 1d); from 2 mm to 5.6 mm (Figure 1e); and the third including both previous groups of particles within the range of 0.5–5.6 mm. These groups were used in the further production of particleboards designated as HS-1, HS-2, HS-12 (Table 1). Given the large volume of leftovers on the 5.6 mm sieve, a mixture of milled fibers and shives was used to produce two exploratory boards (MHF). 

### 2.3. Production

Board samples were manufactured using cold-pressing technology into 200 mm × 200 mm and 400 mm × 400 mm size. Samples were prepared using 266 g hemp shives and a 10 wt.% UF binder for the small pieces and four times the amount for the large ones. Every part of the mixture components was weighed on a scale. Firstly, water was poured into the dry binder and mixed into a homogenous mixture. The stirring was performed for 300 s to obtain a homogenous consistency. Secondly, it was gradually added to the shives while simultaneously mixed with an electric mixer for another 300 s. Hemp shive and binder mixture was formed by hand in the pre-made template (Figure 2a). The template and punch form were immersed in a hydraulic press (Figure 2b) and pressed for 24 h at 21 ± 1 °C under 0.72 MPa pressure for large samples and 2.86 MPa pressure for small ones.

Based on the moisture determination standard LVS EN 322 [54], the water moisture content in board materials is determined using the mass method. 

Immediately after pressing, the samples were weighed and kept under laboratory conditions for 14 days, observing the change in weight using Kern EMB 600-2 digital scale (Kern & Sohn GmbH, Balingen, Germany) with accuracy of 0.01 g at the same time each day.

### 2.4. Density

#### 2.4.1. Cutting and Measuring

Boards of size 400 mm × 400 mm were cut into smaller pieces to determine their structural properties and to prepare them for mechanical and physical testing. Cutting was performed by using a FELDER K 700 S sliding table panel saw (Felder Group, Hall in Tirol, Austria).

A cutting scheme was determined and applied so that the test pieces were accommodated for planned tests according to requirements in each respective standard [60,61,62]. (Figure 3a). A 300 mm × 300 mm piece was used for the thermal conductivity test, a 50 mm × 400 mm piece for bending strength, and 50 mm × 50 mm pieces for the water absorption test.

Test pieces were measured with a digital scale for weight with precision to 0.01 g and with a digital caliper for size with precision to 0.01 mm at each side of the board (Figure 3b). The density was calculated according to LVS EN 323:2000 standard using Equation (2):(2)$$\rho_{w}=\frac{m_{w}}{a_{w}\times b_{w}\times l_{w}}=\frac{m_{w}}{V_{w}}$$

$\rho_{w}$—board density at standard humidity, kg/m^3^. 

$m_{w}$—mass of the test specimen, g. 

$a_{w}$, $b_{w}$—length and width of the test specimen, mm.

$l_{w}$—thickness of the test specimen, mm.

$V_{w}$—volume, m^3^.

#### 2.4.2. Structural Analysis

A Bresser LCD optical magnification microscope was used to analyze the structural uniformity and particle placement within boards. At least five board samples from each produced group were observed under 15× optical magnification on their surface, crosscut view, and surface/side edge where the cut was made to observe dropouts. Measurements for open-air pockets on the surface view were made. 

### 2.5. Water Absorption

To determine the samples most durable to effects of water and moisture, a water absorption test was performed according to LVS EN 317 standard [61] using water with 7 ± 1 pH level and 20 ± 1 °C by immersing 50 mm × 50 mm board samples into water. Board thickness and weight were measured, and absorption amount was calculated according to formula: (3)$$G_{t}=\frac{t_{2}-t_{1}}{t_{1}}\times 100$$
$G_{t}$—swelling in thickness, %. $t_{1}$—thickness of the test piece before immersion, mm. $t_{2}$—thickness of the test piece after immersion, mm. 

Measurements for swelling in thickness and increase in weight were obtained after set periods of time after the immersion for each sample: 15, 30, 45 min, 1, 2, 3, 6, 9, 12, 24, 36, 48 h, 2–10 days. Fourteen groups of samples were subjected to test with three samples within each group. 

The period until total structural disintegration was compared between sample groups to determine the best hemp shive and additive combination.

### 2.6. Bending Strength

Bending strength properties were measured with a FORMTEST UBP 86/200 universal testing machine (FORM+TEST Seidner & Co. GmbH, Riedlingen, Germany) according to LVS EN 310 standard [60]. Load properties were measured on 50 mm × 400 mm size board samples. The length of samples was reduced to accommodate the production and testing capabilities of the machine. To calculate bending strength values from load measurements, this Equation (4) was used:(4)$$\sigma_{b}=\frac{3\;F_{max}\;l_{1}}{2\;b\;d^{2}}$$
$\sigma_{b}$—bending strength, MPa. $F_{max}$—maximum load, N. $l_{1}$—distance between the centers of the supports, mm. b—width of the test piece, mm. d—thickness of the test piece, mm.

For this test, a total of 20 samples from both fraction size boards HS-1 and HS-2 were prepared and crushed for maximum load read and collapse point observation. Maximum load value was used for comparison with other materials; the read was taken from the testing machine. The collapse point was measured from the endpoints of the test piece to determine a possible offset, which means that pressing stage material is unevenly distributed. 

### 2.7. Thermal Conductivity

To compare the developed material with similar chip board materials, it was deemed a thermal conductivity coefficient should be measured. That would give an indication if it were feasible to use the material as part of the insulation structure in the building industry. It was performed with a NETZSCH HFM 446 Lambda heat flow meter (NETZSCH-Gerätebau GmbH, Selb, Germany) according to LVS ISO 8302:2001 [62] by testing 300 mm × 300 mm board pieces at user-defined temperature difference between 0 °C and 20 °C. Thermal conductivity was calculated according to Equation (5):(5)$$\lambda =\frac{Q}{A}\times \frac{L}{\Delta T}$$
λ—thermal conductivity, W/(mK). Q—amount of heat transferred through the material, W. L—distance between the two isothermal planes, m. A—area of the surface, m^2^. ∆T—temperature difference, K.

The results were compared to other recently developed materials with similar density values or similar binding components. All statistical calculations, experimental design, and processing of test results were performed with the Data Analysis Tool Pak software in Microsoft Office 365.

## 3. Results

The production of the hemp board samples went according to the experiment plan (Table 1) and pressure conditions (Table 2). Each obtained sample was useful to acquire new knowledge; even the imperfect samples with surface crumbling or exceeding thickness variation were observed to improve the manufacturing technology.

A total of 40 board samples were prepared. Mean thickness and density values of board types displayed in Table 2 will allow for tracking the particle size range and pressing pressure impacts to determine the combinations that allow making the most of all seed hemp stems in the industry.

### 3.1. Material Properties

The average moisture content in used hemp shives filler material was found to be 10.67 wt.% estimated with a relative error of 1.3% and was taken into account when forming the mixture for molding.

From a group of colorants, acrylic-based solvents showed the best binding capabilities and color equitability compared to ZnO nanoparticle powder and oil-based products. The latter produced boards with dropouts on the surface due to a bad reaction with the binding agent (Figure 4h). A potential use of mineral pigments to tone material was abandoned early because the produced boards started to crumble just minutes after removal from the press and matrix. Water, acrylic, and alcohol-based additives allowed us to create boards with a flat, smooth, and lightly tinted surface (Figure 4a–d). Boards without additives left more openings on their surfaces for air pockets. Although their surfaces were smooth, they gave a rougher feel to the touch (Figure 4e,f). The roughest surface out of samples was observed for MHF boards that were only partially filled with shives (Figure 4g).

When compared, dry pigments fared worse than pre-mixed and soluble pigments due to the additional water necessary to bond with hemp. It was also noted that, while both acrylic and water-based products did similarly well to tone the hemp (Figure 4a,b) the latter needed only ½ of mass to reach the same effect. In fact, the more liquid state the pigment was, the better it bonded with the hemp (Figure 4d). Unfortunately, it contributed to additional water mass that had to be dried out.

A proposition to add the additives to hemp shives beforehand and then dry them out was made and performed. It contributed to more uniformly toned shives as the mixing process could be lengthier. Boards were more even in thickness and dried quickly after removing from the matrix. As the overall moisture content was lower, deformations such as warping and change in thickness during the drying process were smaller. The main problem with this approach was that it lengthened the production process. It was deemed that 7 days of drying the toned hemp shives was necessary to continue with pressing a board sample. 

The results of the produced boards with additives were not conclusive, so it was judged that further evaluation should be addressed in separate research. Further, a possibility to perform an internal bond test to board samples without and with additives should be addressed because of the unknown behavior of the UF binder and pigment composition in the case of simultaneous mixing. This would be the best test to assess binding.

During the experiments and moisture content monitoring, a trend was noticed that during the first 4 days of drying in laboratory conditions (21 ± 1 °C), the samples lost most of their moisture, and during this period, the moister samples (with special additives) started warping. It was also observed that after 10 days of drying, the samples reached a point after which the moisture content change stabilized, and samples were ready for further evaluation and testing.

### 3.2. Structural Analysis

Evaluating the structure of the samples under an optical microscope at 15× magnification, which can be seen in Table 3, a difference can be observed between the groups of samples—the largest size particles (HS-2, HS-2A, and MHF) are arranged more compactly than the smaller particle size samples (HS-1 and HS-1A). In the surface view, the large particles in HS-2 and HS-2A form few large air pockets (average 0.35 mm), while the smaller fractions—HS-1 and HS-2A form many but small air pockets (average 0.15 mm). The smallest air vents were observed in the densest samples—MHF, which consists of dense hemp fibers mesh (86 wt.%) containing relatively few large-size shives particles (14 wt.%). This arrangement directly affects both water absorption rates and thermal conductivity. Further, a lot of dropouts were produced during the cutting process for the smaller fraction (Table 3, column (c), HS-1 and HS-1A). The coarser particles are better bonded to each other and so are rather cut than pulled out by the sawblade.

At the sawn cross sections in (Table 3, column (b)), there is a visible difference in how deeply the additive has tinted the hemp shives—in the case of HS-1A, the tone appears on both the surface and cross section view, while in the case of HS-2A, the color is barely visible in the cross section. This is due to the nature of the additives used, where HS-1A was toned with a product that seeps in and HS-2A with a product producing more of a coating, so only the surface of the particles is tinted.

Significant differences between particle size groups were observed when samples were cut into smaller pieces. Even with the use of an 80-tooth sawblade that is designed for accurate crosscutting, a lot of dropouts for the HS-1 and HS-1A were produced (Table 3, column (c)). As this was not the case for HS-2 and HS-2A samples, it can be clearly stated that boards with larger particle sizes will be easier to handle in post-production processes.

### 3.3. Water Absorption

To evaluate the water absorption, three board pieces for each sample group were soaked in water until total decay occurred. The water absorption test showed that the particle size and density of the samples both had an influence on their longevity. As shown in Figure 5, samples with the same density (285 ± 20 kg/m^3^) but with larger particles were structurally intact for a longer period of immersion—3 days more on average (Figure 5, HS-2 compared to HS-1). Samples with the same particle size in both groups but with higher density endured longer—3 days on average (Figure 5, HS-1A compared to HS-1). Further, the use of shives pre-treatment could improve the resistance to water; for sample groups with the same particle size and density, the water-based wood coating improved the endurance for an extra day (Figure 5—HS-2A) and was still structurally intact and could have endured even longer. The experiment ended after 10 days due to an increase in the growth of mold on the surface of the water and clear benchmark results for comparison.

It was observed that after immersion in water, some particles started to crumble right away and float in the water, thus resulting in sample volume decrease. For the most brittle samples (Figure 5, HS-1A (oil)), extensive surface crumbling started just a few hours after immersion, and then after 48 test hours, they already disbanded. The board samples without additional additives (HS-1, HS-2) lasted longer—the first particles started to crumble off 24 h into the test, but total disbandment occurred after 144 h (6 days) of the test. The least surface crumbling despite internal disbandment was observed for samples with additional additives (HS-1A, HS-2A). 

Most water absorption occurs during the first 5 min of material immersion. After 15 min the board samples without shives pre-treatment and particles size range 0.5–2 mm (HS-1) increased 2.9 times (Figure 6a). Samples with additives (HS-1A, HS-2A) after the same time increased their mass only 2 and 1.4 times, respectively (Figure 6a). After 8–12 h of immersion, the weight gain of samples slows down and later increases gradually (Figure 6b) but does not stop until they start to sink in the soaking container (5–6 days) or totally decay. At the last comparable data point—144 h of soaking in water, the most increase in mass is for HS-1 samples—390%, the least for HS-2A—272%.

Swelling in thickness correlates with the amount of water mass absorbed, with the difference being that hemp fibers (MHF) increase in thickness a lot more than shive-filled boards; the difference is about 50% (Figure 7b—MHF and HS-1). A trend that smaller particles swell 20% more in size than larger particles is observed, as shown by a comparison of HS-1 and HS-2 lines. Further, swelling in thickness at 24 h is reduced by 30–40% for samples with shives pre-treatment in contrast to raw samples with the same particle size and density as evidenced by two pairs of curves—HS-2 and HS-2a, as well as HS-1 and HS-1A (Figure 7b). In addition, the coating effect on swelling thickness is more pronounced in samples with finer particles.

The difference in Figure 6 and Figure 7 shows that while the increase in mass is still happening (more water is still absorbed), the swelling of material slows just after 2–3 h and stabilizes at 24 h, later fluctuating in the range of 5–10%. The fluctuation of swelling data could be caused by the crumbling of surface and bloated forms of the samples. While measuring the soaked samples with a caliper, it is easy to squeeze them and cause a faulty reading.

Research into possibilities of physical or chemical treatment of hemp shives before pressing them into boards should be performed because, in similar experiments around the world, a clear advantage of water absorption reduction has been observed [42,43].

### 3.4. Bending Strength

The bending strength was measured in 20 samples with a density in the range from 261 kg/m^3^ to 297 kg/m^3^ (Figure 8). For 90% of the samples, decay occurred in the center of the sample, which meant they were even in their structure. In the other 10%, the collapsing point offset was up to 10 mm, and they are excluded from further analysis. 

If the particleboard density with coarse shives (HS-2) increases by 14%, bending strength decreases by 19% (Figure 8, **, Equation (7)). At the same time, the increase in the particleboard density from finer shives (HS-1) by 10% results in a bending strength rising proportionally (R^2^ = 0.68), increasing by ~60% when a density reaches 290 kg/m^3^. (Figure 8, *; Equation (6)). Such relationships suggest that due to smaller shive sizes, the total surface area in which contact with the matrix occurs increases, ensuring better adhesion with the binder. This experimental fact requires more in-depth research in the future—if the binder has not been evenly distributed during the pressing process and therefore even denser samples have proved to be more fragile, to find out whether this is really the case, a new series of experiments should be performed, applying the experience gained and ensuring the best possible mixing within the matrix.

The average value calculated from all measurements (Figure 9) estimated with a relative error of 11% was 2.5 ± 0.276 MPa. For the larger particle size group HS-2 boards, it was 2.46 ± 0.228 MPa, and for the smaller particle size group HS-1 boards, it was 2.55 ± 0.326 MPa.
(6)$$\sigma_{b\_avr}*=0.044\;x-9.554$$
(7)$$\sigma_{b\_avr}**\;=-0.012\;x+5.806$$

The bending strength results are close to expectations raised by similar research where UF is used as the binder. In comparison with similar materials (Figure 9), the experimental board average bending strength value is higher than for materials with the same filler, similar density but different binder used [45,63], but is lesser or similar to materials with the higher density [21,33]. There are too many variables, such as the amount and type of binder, particle size, thickness, pressing conditions, and more, to make direct conclusions. Result similarity to denser compositions could be explained by the removal of finer particles (smaller than 0.5 mm) during the material processing step, although another batch of samples with these finer particles included in the study should be produced. To match denser materials such as wood particle boards, wood chipboards, and medium density fiberboards, the bending strength values still need to be improved.

### 3.5. Thermal Conductivity

The thermal conductivity coefficient was measured in eight board samples with broad density properties. In practice, the thermal conductivity λ does not depend on the bulk density of the sample ρ varying from 205 to 316 kg/m^3^ (by 35%), variance in thickness, and particle size groups. The differences in λ-values shown in Figure 10 are statistically insignificant. The average value calculated from the 8 λ measurements (Figure 10), estimated with a relative error of 3.7%, is: λ = 0.05703 ± 0.00213 W/(mK). To research the thermal conductivity properties in more detail, a larger number of samples should be subjected to testing.

The differences in the conductivity values seen in Figure 10 have originated mainly from the particle size range and binder distribution. In compared materials, the type of binders used was formaldehyde resin glue. It seems that thermal conductivity disperses heavily if the range of particle sizes is too wide (Figure 10, black triangles), which suggests an unstable board structure. In this research, the particles smaller than 0.5 mm were removed to both improve the binding process and leave clear air pockets, which are important to slow down the transfer of heat flow. In future research, hemp shives board samples that have the same particle size ranges but different binders must be produced to determine specific influence. In comparison with other hemp shives and wood chipboard materials with similar density (Figure 11), experimental samples have a lower thermal conductivity.

## 4. Discussion

Through the production of board samples, it was deemed that separating the process of additive application and using a binding agent produces better results but consumes more time. Additionally, additives with color pigments allow producing better-looking textures, which was an unintended benefit to the material.

A potential problem for manufacturing such board material is the time needed to dry the boards. With cold pressing technology, the energy resources that would be used to apply heat are saved, but the process is lengthier. To optimize the use of resources and manufacturing, a possible hybrid method should be evaluated because the binder is applicable with a heating technology where at a 60 °C temperature, the drying time is only 6 min compared to 270 min without heat application.

To fully address the pros and cons of the fiber material, an evaluation of additional properties should be performed—modulus of elasticity, internal bond strength, screw withdrawal tests are worth assessing as these are important properties for proper material application in the industry. Assessment of acoustic transmission should be evaluated as well as it is an intriguing and sometimes overlooked aspect of insulation materials in general. As the water absorption test showed, a more precise investigation into mold growth is needed to understand the microbial resistance of hemp.

Future research directions include proposed material production with a more ecological binder such as lignin (magnesium or sodium lignosulfonates) and a comparison between binding agents in specific particle size groups. For this, a calculation of environmental impact and production cost for each produced board type should be performed. This evaluation would move produced material towards valorization. In between still research into the influence of particle size should be performed determining different particle size groups and investigating the potential of milled fiber use in board production. The mechanical properties could be significantly improved by lamination with technical fabrics or fibers.

## 5. Conclusions

A successful set of hemp shives board samples was produced during this experiment. It was confirmed that reducing the original particle size group of 0.063 mm to 15 mm shives to particle size group of 0.5 mm to 5.6 mm shives using a hammer-type mill and sieving produces a uniform material for board production. The acquired particle size is relevant to other materials produced from organic particles such as wood chips. An affirmation of chosen cold pressing technology and mold construction was reached to solidify an approach that will be used in future research—a total of 40 board samples were produced with this technology. 

The determined particle size groups that were used in board production were acquired by sieving the milled hemp and consisted of 0.5 mm to 2 mm shives in the first group and of 2 mm to 5.6 mm shives in the second group as well mixed fiber-shive boards allowing the use of 90% of the mass of dry seed hemp (*Canabis Sativa*) crushed stems in three types of boards differing in properties. This could be a sustainable solution to increase the added value of the hemp seeds acquisition by-product that has been used still now with low efficiency.

Leftovers of milled fibers of size over 5.6 mm in this study were used to produce two boards for indicative testing of water absorption and structural analysis. Both main groups showed little difference when compared in equal density boards. The larger particle size group boards consisting of 2 mm to 5.6 mm hemp shives had fewer particle dropouts on surfaces of the board, sawn cross sections, and edges; they fared better on both bending strength and water absorption tests. This indicates that these boards are better suited for post-production processes such as sawing and routing when boards will be prepared for application in projects. 

The water absorption test confirmed that the chosen type of binder decays swiftly in water, and hemp shives soak up a lot of water. The longevity of board samples can be improved by using a larger particle size (6 days until total decay for 0.5 mm to 2 mm size shives to 9 days for 2 mm to 5.6 mm size shives) and further improved by adding additives such as water base coating to the shives before pressing the board, making it last until the experiment was finished due to extensive mold growth on the surface of the water on day 10. Thickness swelling was 20% lower for boards with a larger shives group and further 35% lower with additive—water base coating. Observed characteristics point to possible use in environments with little direct contact with water and moisture, such as interior projects.

Bending strength variation between particle size groups was only 4%, but the decreasing trend line of bending strength value change in response to minimal changes in density for particle size group from 2 mm to 5.6 mm indicates a possible uneven distribution of the binder in the prepared board samples. The average value for 18 measurements was 2.5 ± 0.276 MPa. This is a significantly lower bending strength value than particleboard from wood chips or boards with different binder possess. With such low bending strength, the boards must be used together with a frame of another material or layer of laminate that supports the vertical load in building constructions. Boards can be used as non-load-bearing applications such as part of an interior room dividing wall.

The thermal conductivity value for the produced board material was 0.05703 ± 0.00213 W/(mK), which is better than similar materials in the developmental or production stage provided. It was observed that particle size has an insignificant influence on thermal conductivity properties. Further exploration into variance between different particle size groups should be investigated in future research. At this stage, the produced material should be viewed as an alternative to fiberglass and foam insulation materials—one that has less impact on the environment during manufacturing. It can be used as an environmentally friendly solution for insulation in passive houses.

## Figures and Tables

**Figure 1 materials-15-00886-f001:**
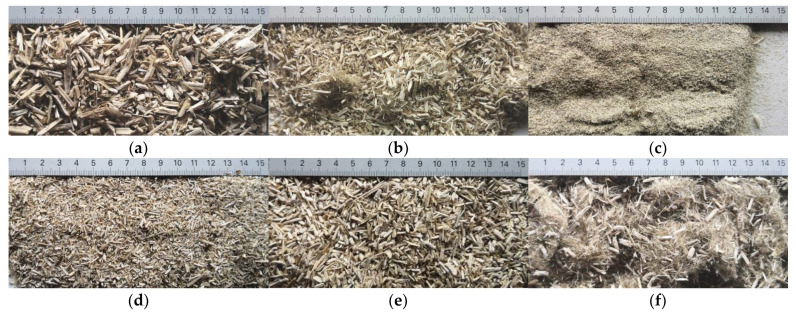
(**a**) HS before processing; (**b**) HS after milling; (**c**) HS particle size less than 0.5 mm; (**d**) HS-1 particle size range (0.5–2 mm); (**e**) HS-2 particle size range (2–5.6 mm); (**f**) MHF particle sizes over 5.6 mm.

**Figure 2 materials-15-00886-f002:**
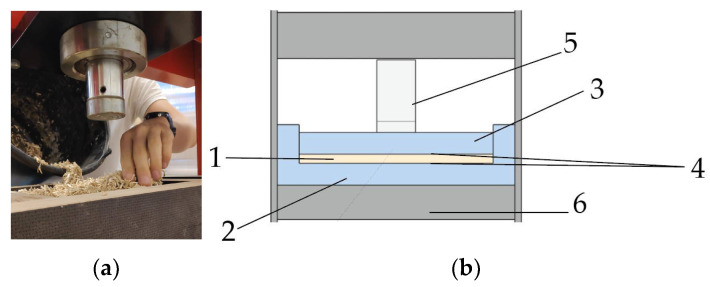
(**a**) Formation of mixture into the mold; (**b**) forming setup of hydraulic press and mold: 1—Hemp shive mass that is formed into board; 2—bottom part of the mold; 3—punch; 4—layers of non-stick paper; 5—hydraulic jack, 6—hydraulic press stand.

**Figure 3 materials-15-00886-f003:**
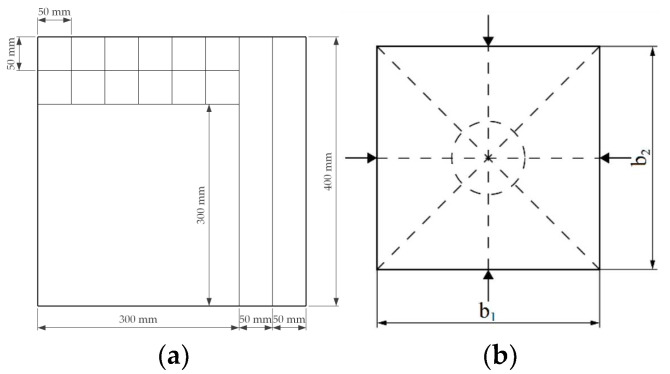
(**a**) Cutting scheme for boards produced in the study. (**b**) Scheme for determining the thickness of the samples.

**Figure 4 materials-15-00886-f004:**
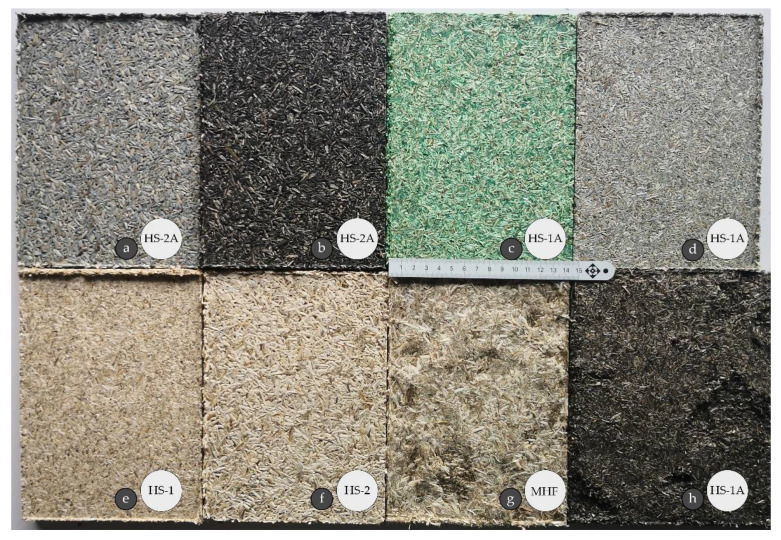
Surface view of board samples: (**a**) HS-2A sample with water-based wood coating applied on HS; (**b**) HS-2A sample with acrylic colorant applied during mixing process; (**c**) HS-1A sample with oil-based coating applied on HS; (**d**) HS-1A sample with water-based wood coating applied during mixing process; (**e**) HS-1 board sample; (**f**) HS-2 board sample; (**g**) MHF board sample; (**h**) HS-1A with oil-based coating applied on HS.

**Figure 5 materials-15-00886-f005:**
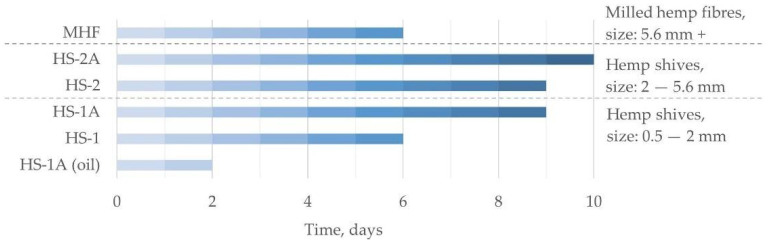
Amount of time until hemp board samples decay in water.

**Figure 6 materials-15-00886-f006:**
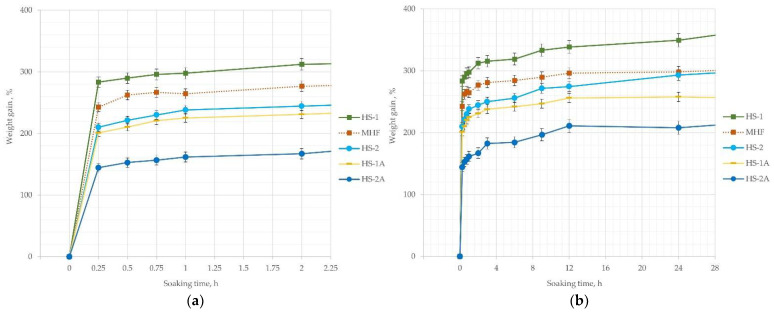
Water absorption—increase in mass of hemp board samples: (**a**) weight gain in 2 h; (**b**) weight gain in 24 h.

**Figure 7 materials-15-00886-f007:**
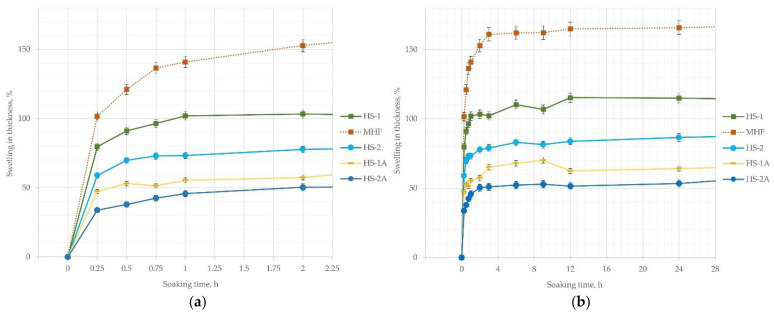
Water absorption—swelling in thickness of hemp board samples: (**a**) thickness swelling for 2 h; (**b**) thickness swelling for 24 h.

**Figure 8 materials-15-00886-f008:**
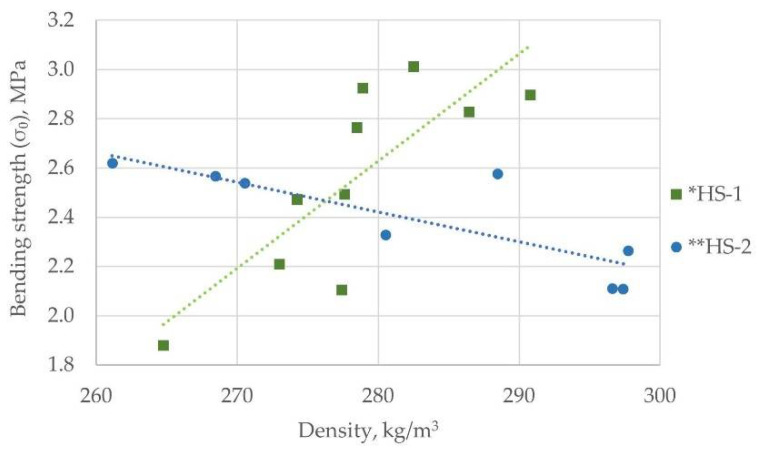
Bending strength comparison for hemp shives boards from both particle size groups.

**Figure 9 materials-15-00886-f009:**
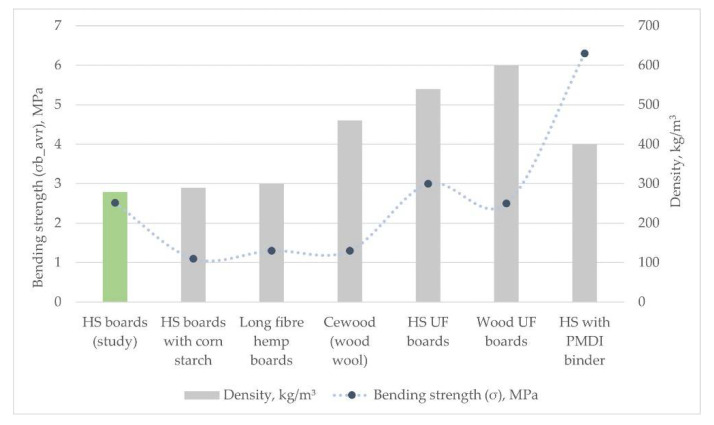
Bending strength comparison of hemp board samples produced in the study to other materials consisting of hemp or wood particles [21,27,33,45,48,63].

**Figure 10 materials-15-00886-f010:**
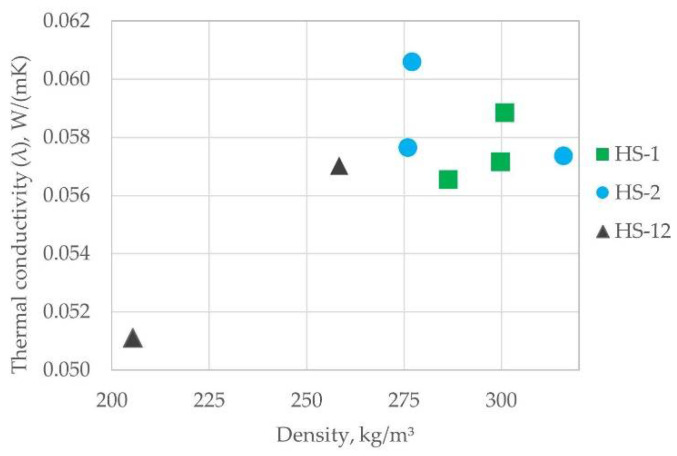
Thermal conductivity values for hemp shive samples produced in the study.

**Figure 11 materials-15-00886-f011:**
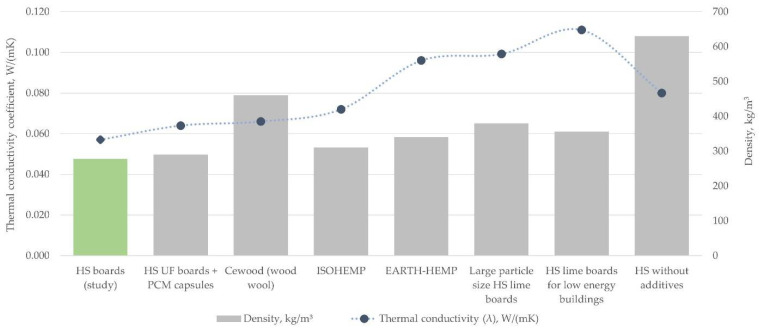
Thermal conductivity of hemp shive board samples in comparison to similar materials consisting of hemp or wood particles [47,48,49,50,53,64,65].

**Table 1 materials-15-00886-t001:** Classification of the hemp shive boards manufactured in the study.

Type	Particle Size (mm)	Pre-Treatment/Additives						Tests Conducted *			
		Not Applicated	Oil-Based Coating	Water-Based Wood Coating	Acrylic Colorant	Mineral Pigment	Zinc Oxide Powder	BS	SA	TC	WA
HS-1	0.5–2	X						X	X	X	X
HS-2	2–5.6	X						X	X	X	X
HS-12	0.5–5.6	X								X	
HS-1A	0.5–2		X/-	X/-		-/X			X		X
HS-2A	2–5.6		X/-	X/-	-/X		-/X		X		X
MHF	5.6+	X									X

* Explanation for abbreviation: SA—structural analysis, WA—water absorption, BS—bending strength, TC—thermal conductivity.

**Table 2 materials-15-00886-t002:** Produced hemp shive particleboard mean values of the physical properties.

Type	Pressure Conditions (MPa)	Thickness (mm)		Density (kg/m^3^)	
	Mean	Mean	Confidence Level	Mean	Confidence Level
HS-1	0.72	20.7	0.9	285	9
HS-2	0.72	21.2	1.1	283	12
HS-3	0.72	26.7	2.2	230	23
HS-1A	2.86	17.8	0.7	373	16
HS-2A	2.86	18.7	0.8	367	13
MHF	2.86	16.3	2.8	545	10

**Table 3 materials-15-00886-t003:** Material structure analysis at 15× magnification.

Type	Density (kg/m^3^)	Particle Size Ranges (mm)	Surface Structure (a)	Sawn Cross Sections (b)	Sawn Edge (c)
HS-1	275	0.5–2	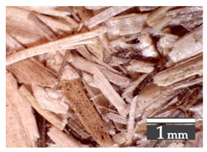	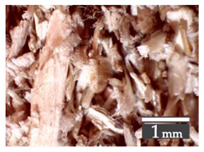	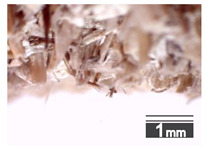
HS-2	283	2–5.6	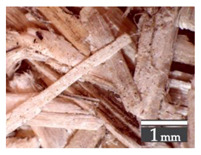	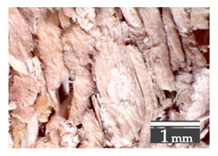	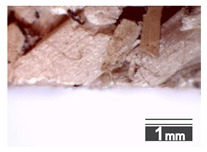
MHF (86% fibers, 14% shives)	545	5.6+	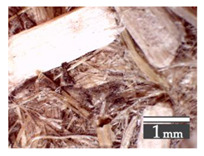	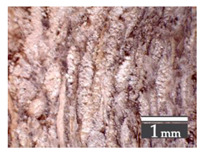	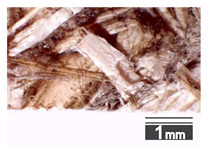
HS-2A	378	2–5.6	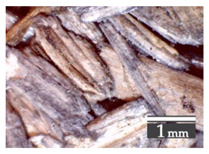	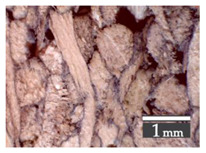	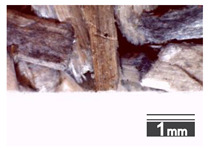
HS-1A	381	0.5–2	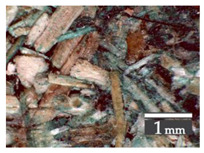	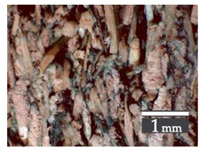	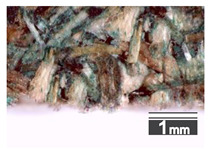

## Data Availability

Not applicable.

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
