# Peer review of "Production of Particleboard Using Various Particle Size Hemp Shives as Filler"

_materials, 2022, doi:10.3390/ma15030886_

Round 1
Reviewer 1 Report
Review report of the manuscript materials-1515161 "PRODUCTION OF PARTICLEBOARD USING VARIOUS PARTICLE SIZE HEMP SHIVES AS FILLER".
The topic of the manuscript is interesting, the results are novel. The research idea is clear. There are some issues, that needs to be fixed before publication.
Comments:
Line 9: What do you mean by "...is doing careful research".
Lines 9-11: These sentences are too general, please consider omitting them.
Line 12: Why exactly hemp shives?
Line 14: Size from 500 mic to 5 mm - please check the Materials and methods section, there are different sizes.
Line 15: "resin glue" please reformulate.
Line 20: Thermal conductivity instead of heat conductivity, water absorption (without amount).
The abstract: Please add some results, discuss also the novelty of your research, implications for practice.
The Introduction:
This part is insufficient. Please discuss previous research in this area, please check for example here:
https://hrcak.srce.hr/29389
https://ojs.cnr.ncsu.edu/index.php/BioRes/article/view/11788
and many others.
Please add also general information about PB production (in 2018 97 million m3 worldwide): doi.org/10.3390/ma14174875
Please discuss also how a problem with hemp absorption capacity influence the TS and WA compared to wood residues.
Please in the Introduction part discuss also size of particles, what about previous research in this area, how the particle samples influence the properties of the PB? Please check for example here:
https://www.researchgate.net/profile/Ismail-Oezluesoylu/publication/329674033_The_Effect_of_Chip_Size_on_the_Particleboard_Properties/links/5c14e62092851c39ebee79b1/The-Effect-of-Chip-Size-on-the-Particleboard-Properties.pdf
https://www.mdpi.com/2073-4360/13/7/1125
Please add also a discussion about the thermal conductivity of PB and other types of boards.
What about the resin? Why UF? Please add some sentences about that.
Line 66: The size does not correspond with the Abstract.
Please add more information about UF resin.
Please add research design of adding additives, amounts, phase of the production, etc.
Line 88: Please check the template for references [22-25]
Line 93-96: Please check again the size of particles.
Lines 100-103: Please be more specific about these procedures.
Line 108-120: Please use was instead is.
Line 134: No need to add Equation 2, reference for the standard is enough, same for other properties. The tests are well known. Same for Figure 4.
What about the Thickness swelling? After 2 and 24 hours?
What about MOE and IB? Why only MOR was examined?
Statistical analysis of data?
The Materials and methods part is insufficient, please be more precise, check for example here:
doi.org/10.3390/ma14174875
The results and discussion:
Please discuss the statistical significance of the results.
Chapter 3.5 Please use thermal instead of heat conductivity.
Please add more discussion with other research in the case of TC (line 301-302), please discuss differences and causes.
In the conclusions part please discuss the novelty of your research, limitations and implications for practice.
Results and discussion:
Please discuss how particle size influence Thermal conductivity, check for example here:
doi.org/10.3390/polym13142287
Author Response
All the requests for revisions have been taken seriously and the article has been heavily expanded and revised. All the responses to each of the points are written in the attached document.

Reviewer 2 Report
Dear Authors
I have reviewed your manuscript submitted to the journal Materials. The general idea is interesting and certainly in line with the current trend of finding new solutions for structural materials based on renewable sources, especially plants. However, your work needs considerable improvement in order to be published as a scientific paper. Below are some suggestions for necessary corrections and additions:
1. the Introduction should be expanded. Reference should be made to similar applications in particleboard production that are based on the use of other plants, plant waste (e.g. coffee grounds) and especially wood chips.
It should be explained more precisely why hemp is such a unique material. In this discussion, the idea should be supported with appropriate references.
(2) Most of the tests described lack specific information on how many times a given measurement was repeated.
3. the results need to be presented more clearly. All graphs should show the standard deviation for a given measurement (we assume that there were several repetitions of a given measurement).
4. graphs 7 and 8 have a badly described horizontal axis, which is named as measurement points. In the text they are referred to as minutes.
From graphs 7 and 8 it can be seen that the tested materials increased their mass and thickness by 100 - 250 % after only 1 minute. Is this an error in the graph or the real data? If true, this phenomenon has not been discussed.
5. there is no statistical analysis of significance of differences in properties of analysed materials
6. the conclusions require rewording. They should focus on highlighting the obtained results supported by statistical analysis. e.g. "this has higher values for these properties and this is probably because...". Additionally, future directions should be presented.
Author Response

(The authors gave the same response as above.)

Reviewer 3 Report
The manuscript deals with the hemp boards by cold pressing method. The board density, structural properties, bending strength, heat conductivity and water absorption amount were discussed and compared with similar materials already in industry or in development stage. However, the manuscript should highlight its innovation, and different parameters on the properties of hemp boards should be described in detail. Below I am adding my comments.
|
1. The authors prepared hemp boards by cold pressing method, using hemp and urea formaldehyde resin glue as the raw materials. The novelty of this work should be emphasized in introduction and the properties should be compared with other reports.
2. Cutting and measuring: 300 x 300 mm piece for hear conductivity test, a 50 x 400 mm piece for bending strength and 50 x 50 mm pieces for water absorption. Why did the authors choose different sizes for different tests? Are there any test standards to refer?
3. A total of 40 board samples were pressed with different particle sizes, densities, added additives. The authors should list all/optimized technological parameters and performance parameters of these samples.
4. The authors should compare the obtained results in discussion section, and explain the effect structure, particle size, density and additives on the properties of hemp boards.
|
Author Response
All the requests for revisions have been taken seriously and the article has been heavily expanded and revised. All the responses to each of the points are written in the attached word document.
I wish to thank you for clearly describing the specific revisions and improvements necessary for the article.

Round 2
Reviewer 1 Report
The manuscript was significantly improved.
Author Response
Thank you for taking the time to read and comment on this article. The comments provided help to improve the quality of the article.
In this version of the revision, the units used throughout the article were arranged correctly and some sentences were reworded in the results section to improve the readability of the article.
Spellchecking was done to correct all typographical errors.
Reviewer 2 Report
Dear Authors,
thank you for your responses to my remarks and comments on your manuscript. Overall, the article as it stands, with the changes and additions made, looks good, from a scientific and visual point of view. However, I still have some minor comments that I think should be taken into account before publication.
# line 395 - wrong density unit (upper intercept should be used);
# lines 402-405 - wrong unit of bending strength, should be MPa;
# figure 8 - wrong units of density and bending strength;
# figure 9 - wrong units of density and flexural strength (not consistent with text where MPa is used). Additionally, there is an incorrectness in the figure related to bending strength. The text on lines 409-412 refers to the average values of bending strength. The figure shows the parameter Fmax, which suggests a maximum value. Moreover, the symbol F in the ISO standard stands for force, not stress/strength. For stress, the symbol σb_avr (bending strength) should be used, with an index indicating the average values;
# Figures 10 and 11 - wrong density units
Author Response
Thank you for your time in reviewing the post and for your detailed comments during the review process.
Throughout the article, the units for density, thermal conductivity, dimensions (millimeters, microns) were checked and corrected. Also the order of the tables and according references in the text was corrected.
Bending strength (flexural strength) symbol has been corrected in both the methodological part and the result part according to standard and your review. I acknowledge the misuse of Fmax symbol in the graph (line 392) and equation 4 (line 250), Both have been corrected.
Minor rewording and clarification in the results section has been done (mostly specifying references in text to the graphs that are explained). All changes have been done with Track Changes option.
Reviewer 3 Report
Your paper can be accepted conditionally accepted.
Author Response
Thank you for taking the time to read and comment on this article. The comments from you provided necessary help to improve the quality of the article during revision process.
In this version of the revisions, the units used throughout the article for density, thermal conductivity, size were arranged correctly and some sentences were reworded in the results section to improve the readability of the article and understanding of results.
Spellchecking was done to correct all typographical errors.